# Training Physicians in Motivational Communication to Address Influenza Vaccine Hesitation: A Proof-of-Concept Study

**DOI:** 10.3390/vaccines10020143

**Published:** 2022-01-19

**Authors:** Sara Labbé, Inés Colmegna, Valeria Valerio, Vincent Gosselin Boucher, Sandra Peláez, Anda I. Dragomir, Catherine Laurin, Elizabeth M. Hazel, Simon L. Bacon, Kim L. Lavoie

**Affiliations:** 1Montréal Behavioural Medicine Centre, Centre Intégré Universitaire de Santé et de Services Sociaux du Nord-de-l’Île-de-Montréal (CIUSSS-NIM), Hôpital du Sacré-Coeur de Montréal, Montreal, QC H4J 1C5, Canada; labbe.sara@courrier.uqam.ca (S.L.); vincent.gosselin.boucher@gmail.com (V.G.B.); anda.dragomir22@gmail.com (A.I.D.); catherinelaurin@yahoo.ca (C.L.); simon.bacon@concordia.ca (S.L.B.); 2Department of Psychology, Université du Québec à Montréal, Montreal, QC H2X 3P2, Canada; 3Division of Rheumatology, Department of Medicine, McGill University, Montreal, QC H3G 1A3, Canada; Ines.Colmegna.med@ssss.gouv.qc.ca (I.C.); elizabeth.hazel@mcgill.ca (E.M.H.); 4The Research Institute of the McGill University Health Centre (MUHC), Montreal, QC H4A 3J1, Canada; valeria.valerioguillen@affiliate.mcgill.ca; 5School of Kinesiology, University of British Columbia, Vancouver, BC V6T 1Z3, Canada; 6School of Kinesiology and Physical Activity Sciences (EKSAP), Faculty of Medicine, University of Montreal, Montreal, QC H3T 1J4, Canada; sandra.pelaez@umontreal.ca; 7Research Center of Sainte-Justine University Hospital (RC-CHUSJ), Montreal, QC H3T 1C5, Canada; 8Department of Exercise Science, Concordia University, Montreal, QC H3G 1M8, Canada

**Keywords:** motivational communication, rheumatoid arthritis, influenza, vaccine hesitation, vaccines

## Abstract

Background: Strategies to support health care professionals on how to address vaccine hesitancy are needed. Methods: We developed a 4-h Motivational Communication (MC) training program tailored to help physicians address hesitancy related to influenza vaccination among patients living with rheumatoid arthritis. Five MC competencies were evaluated at baseline and post-training with a standardized patient using the Motivational Interviewing Treatment Integrity [MITI] scale. Adherence to MC during clinical consultations and changes in vaccine intentions was measured as secondary outcomes. Results: Seven rheumatology physicians participated in the training. MITI scores increased in all participants, and 71% (n = 5) achieved thresholds of clinical competency (i.e., ≥3.5/5 at MITI global score and ≥3/5 on at least 3 individual competency score) post-training. Autonomy/support and empathy competencies reached competency thresholds (+2.4 ± 1.3 to +4.1 ± 0.7 and +2.1 ± 0.7 to +4.1 ± 0.7, respectively). Evocation and collaboration competencies improved but without reaching competency thresholds (+1.4 ± 0.8 to +3.1 ± 1.1; +1.4 ± 0.8 to +2.9 ± 1.1, respectively). Direction did not improve. Among 21 patient consultations post-training, 15 (71%) were MC-consistent. Of the 15 patients, 67% (10/15) intended to receive the influenza vaccine and 33% (5/15) received it. Conclusion: A brief MC training program targeting vaccine hesitancy increased MC competency among rheumatology physicians and promoted behavioral change among patients.

## 1. Introduction

Rheumatoid Arthritis (RA) and its treatment increases the risk of infections and infection-related complications, which are a major cause of premature death in this population [1,2,3,4,5]. Despite this, less than 50% of people living with RA are vaccinated for influenza in the US and Canada [6,7].

Suboptimal vaccination coverage is not exclusive to patients treated with rheumatic diseases by rheumatology or to the influenza vaccine [8]. Vaccination hesitancy is an increasingly important phenomenon now listed among the biggest threats to global health according to the World Health Organization [9,10] and defined as: “… a delay in acceptance or refusal of vaccines despite availability of vaccination services. Vaccine hesitancy is complex and context specific, varying across time, place and vaccines. It is influenced by factors such as complacency, convenience and confidence.”.

Common interventions to increase vaccination uptake include facilitating access to vaccines, sending reminders, and educating patients on the importance of immunization [11,12]. While reminders and educational interventions might be beneficial for patients with favourable vaccine attitudes, they are often insufficient for improving vaccination rates among patients who may be vaccine hesitant [12,13,14]. Indeed, providing unsolicited advice or information about the benefits of adopting specific health behaviors (e.g., vaccination, exercise, smoking cessation, etc.) has been shown to increase patient resistance, suggesting that novel approaches are needed [15]. 

Motivational Interviewing (MI) is an increasingly popular behavior change counselling approach used to motivate and support patients to adopt and maintain specific health behaviors, including vaccination [15]. MI has been shown to be effective in promoting smoking cessation, physical exercise, and medication adherence, as well as reducing alcohol consumption and at-risk sexual behaviors [16,17,18,19]. In the context of vaccination, a study with a quasi-experimental design demonstrated that one session of MI delivered postpartum to mothers resulted in a 15% increase in the intention of mothers to vaccinate their infants and an increase in vaccine coverage in infants [20]. A recent randomized controlled trial (RCT) also demonstrated that 20-min of counseling using MI techniques by a trained research assistant could significantly decrease the proportion of patients who were vaccine hesitant [21]. However, few studies have aimed to train healthcare providers (HCPs), especially physicians, to use MI techniques to address vaccine hesitancy [22,23,24].

The success of MI in promoting health behavior change lies in having knowledge of behavior change processes, as well as expertise in using specific competencies and techniques [25]. Therefore, this approach requires several hours of training as well as extensive supervision [24]. While this might be possible to implement in the context of psychotherapy or counselling training, it may not be feasible for physicians with busy medical practices and limited time to devote to training.

Motivational communication (MC) is an approach recently developed to specifically provide basic behavior change counselling training to HCPs. MC is defined as an evidence-based, time-efficient communication style used by health care providers to promote patient engagement, adoption of healthy behaviors and sustained self-management of chronic conditions. It is informed by the behavioral sciences and emphasizes shared decision-making that is tailored to the patient’s preferences, goals and values [26]. MC was developed collaboratively with HCPs and behavior change experts using an integrated knowledge translation (iKT) approach where all stakeholders were involved in the development process to optimize acceptability and uptake. This collaborative process led to the identification of 11 core communication competencies that are: (1) evidence-based for promoting behavior change, (2) seen as within HCP’s scope of practice, and (3) seen as feasible to incorporate into clinical practice [27]. The competencies were incorporated into a brief 4-h basic training program targeting physicians. 

As part of an ongoing program to enhance influenza vaccine uptake among patients with RA and in preparation for a RCT, we conducted a proof-of-concept study to determine if a 4-h MC training program was ‘strong enough’ to help MC-naïve physicians achieve minimum threshold levels of MC competency. This early phase proof-of-concept study (as defined by the ORBIT Model Framework) was conducted as part of a larger research program that aims to develop and test the efficacy of MC training among physicians, which is consistent with guidelines on behavioural intervention development and testing [28]. We adapted the program to target physicians to specifically address influenza vaccine hesitancy among patients with RA.

## 2. Materials and Methods

### 2.1. Study Design

This study was conducted at the Division of Rheumatology of the Montreal General Hospital, McGill University Health Centre (Montréal, QC, Canada). Rheumatologists and rheumatology residents were invited to participate. Baseline assessment of their MC competencies was done with a validated assessment tool (Motivational Interviewing Treatment Integrity [MITI] scale, see below) [29]. Subsequently, physicians received a 4-h MC training program addressing patients with RA who were influenza vaccine-hesitant. Within a week post-training, physicians underwent an assessment of their MC competencies. MC assessments (baseline and post-training) consisted of a 10-min role-play with a standardized patient with RA who was vaccine hesitant. The role-plays were recorded and assessed using the MITI scale, by an independent rater, a graduate psychology student proficient in MC (AD). The MITI score pre- and post-training was used to evaluate MC competency changes.

MC training occurred in the month of September, prior to the Northern Hemisphere influenza season. This allowed physicians to use their newly acquired MC skills in their vaccine discussions with patients. Prior to these consultations, patients were asked to complete a short demographic questionnaire including rheumatology diagnosis, age, and sex. They were also asked to indicate their intention to receive the influenza vaccine (yes, no, maybe, see below). Patients who demonstrated vaccine-hesitancy were offered participation in the study and were given a copy of the informed consent document. Study participants met with a MC-trained physician and received 5–10 min of vaccine counselling using the MC approach. Immediately after their consultation, patients again answered questions about their intention to be vaccinated. An overview of the study design is presented in Figure 1. 

### 2.2. Participants, Rheumatology Physicians

To be part of the study, rheumatology physicians needed to meet all the following inclusion criteria: (1) rheumatologist or rheumatology resident; (2) active practice including patients with RA; and (3) a baseline MITI score below the minimal score for MC competency (i.e., 3.5). Clinicians were recruited from a single rheumatology division that in September 2019 included 14 rheumatologists and 6 rheumatology residents. Invitations to participate in the MC training were sent via email and through personal contacts between team members. Physicians interested in participating were invited to contact a member of the research team. All participants received the same 4-h MC training program delivered to the entire group by two MC experts. Physicians were not compensated. 

### 2.3. Patients with RA

Patients with RA presenting to the outpatient rheumatology clinic were recruited between October 2019 and January 2020. To be eligible, patients must have: (1) been adults (age ≥ 18); (2) been diagnosed with RA or Juvenile Idiopathic Rheumatoid Arthritis (JIA); (3) been fluent in English or French; (4) had a consultation with a physician trained in MC; (5) answered “no” or “maybe” to the question: I plan to receive the flu vaccine this year; and (6) accepted to discuss the influenza vaccine with their physician. Patients answering “no” were defined as vaccine-refuser and those answering “maybe” were defined as vaccine-hesitant. Patient participants were included in a draw to win one of four Amazon gift cards worth $50.

### 2.4. Interventions and Measures: Clinicians

#### 2.4.1. MC Competency with a Standardized Patient

Motivational Interviewing Treatment Integrity Scale (MITI): MC competency among physicians pre-post training (primary outcome) was evaluated with the MITI, a validated tool designed to measure 5-key competencies of MI common to MC: Evocation, Collaboration, Autonomy/support, Direction, and Empathy [29]. The reliability of the MITI global and subscale scores were estimated using intra-class correlations coefficient (ICC) with three pairs of coders and found them to be “fair” to “excellent”, with 70% of the subscales falling in the “excellent” range. For this study, one independent rater skilled in MC and blind to the research objectives and physician training status (pre or post) rated physician performance on each of the MITI competencies during a 5–10-min audio-taped role-play exercise between the physician and a standardized patient who was vaccine hesitant. This patient used a script informed by a literature review on the determinants of influenza vaccine hesitancy in this population [30]. 

#### 2.4.2. MC Competency in Practice

Within one to 12 weeks post-training, the use of MC skills with patients with RA expressing vaccine hesitancy was assessed among trained physicians with the MITI. A rater skilled in MC assessed audio recordings of consultations, or directly observed consultations (for patients who refused to have their consultation audio-recorded) as described above. 

### 2.5. Patients

#### 2.5.1. Vaccine Hesitancy/Intention

Vaccine hesitancy/intention among patients with RA was assessed using the proportion of patients answering yes (post-intervention only), no and maybe to the question: “I plan to receive the flu vaccine this year”. 

#### 2.5.2. Vaccination Rates

The influenza vaccination rate after the consultation with the MC trained physicians was defined according to the Quebec Vaccination Registry (QVR). This registry is an electronic database that provides HCPs with vaccination records in the province of Québec.

### 2.6. Motivational Communication Training Program

The MC training program was a 4-h face-to-face session. The first hour was a general introduction to MC, followed by a 3-h competencies-building workshop that included four modules: (1) understanding behavior change; (2) identifying the attitudes and core competencies of MC; (3) recognizing how MC can be used to engage patients in their care and overcome resistance to change; and (4) demonstrating the basic competencies of MC [15]. In the last module, participants were taught how to use open questions to elicit change (evocation skills), the importance of reflective listening and expressing empathy, and how to provide information in an autonomy-supportive fashion (i.e., by asking permission, providing information neutrally, and asking for feedback), all using a case-based approach. 

The training was delivered by two senior clinical psychologists with expertise in MC (CL and KL), using six behavior change techniques from Michie’s behavioral taxonomy adapted to group training [31]. First, instruction how to perform behavior (BCT #4.1) consisted of providing participants with the ‘do’s and don’ts’ of using an MC approach. Secondly, the trainer proceeded with demonstrations (BCT #6.1) of MC using role-play. Third, participants were asked to practice (BCT #8.1) MC competencies within small groups. Fourth, participants were asked to self-monitor their behavior (BCT #2.3) by providing a general evaluation of their performance during and after the role-plays. Fifth, immediate feedback on the participant’s behavior (i.e., use of the MC competencies, BCT #2.2) and finally, feedback on outcomes of behavior (BCT #2.7) was provided by the trainer, during and after role-plays. The template for intervention description and replication (TiDieR) checklist is available in Appendix A [32].

### 2.7. Outcomes 

In line with guidelines for establishing efficacy thresholds for proof-of-concept studies [28], we defined success in our primary outcome if >50% (4/7) of physicians reached a minimum global competency levels of 3.5/5 or greater on the MITI global score after MC training. In order to ensure that physicians acquired a range of skills, they also had to achieve a minimum score of three (out of 5) or greater on at least three out of the five competencies assessed by the MITI post-training. 

As secondary outcomes, the physician’s adherence with MC during consultations with patients with RA who were vaccine hesitant, was reported as the % of counselling sessions recorded or observed where the physician’s communication achieved threshold competency levels on the MITI. MC counselling sessions were then classified as “MC-consistent”, or “MC-inconsistent”. Among the MC consistent and the inconsistent consultations, we assessed changes in influenza vaccine intentions from baseline to post-consultation and the proportion of patients vaccinated according to QVR. 

### 2.8. Data Analysis

Given that the objectives of a proof-of-concept study are to determine the extent to which an intervention shows a “ signal” of being able to achieve a pre-specified minimum threshold of impact (i.e., clinically significant threshold) rather than a statistically significant difference between two groups, data was summarized using means (SD) and proportions for continuous and categorical data, respectively. The chi-square test for categorical variables was used to inspect the between-groups differences on vaccine hesitancy changes and vaccination rates using Microsoft Excel 2021 software.

## 3. Results

### 3.1. Participants’ Characteristics

Of 21 eligible physicians, seven (33%, 6 residents and 1 senior rheumatologist) were recruited and completed MC training. All of them were women. The main reason for other rheumatologists’ non-participation was stated as limited time to undergo training.

A total of 112 patients with RA were approached to determine eligibility. Sixty-five patients were excluded because they either planned to receive the influenza vaccine or had already received it. Seven refused to participate and the remaining 40 were eligible. Among those, 7 were excluded due to language. Three patients with whom vaccine was not discussed during the consultation were excluded from the analyses. Thirty patients were included in the final analyses. Most participants were middle-age (mean, SD 36 ± 16.2 years, min: 19, max: 73), female (n = 27, 90%), and had a diagnosis of RA (n = 22, 73%). Four patients self-reported a diagnosis of JIA (13%) and another four patients (13%) were unsure if their diagnosis was RA or JIA. Most patients (n = 21, 70%) consented to have their consultations either audio-recorded (n = 7, 23%) or observed (n = 14, 47%). All 9 patients who did not consent were classified as hesitant to receive the vaccine.

### 3.2. Fidelity of Training Delivery

A member of the research team (SL) observed the training sessions and completed a checklist to assess training fidelity. During the workshop, all 4 modules were covered and all communication strategies were taught (using open questions, reflective listening, providing information) using a case-based approach, as intended. Finally, all six BCTs described in the methods were used to deliver the training session, as intended.

### 3.3. Primary Outcome

After the MC training, all seven physicians improved their MITI score (Figure 2a). Most participants (5/7, 71%) achieved clinical competency (i.e., global MITI scores ≥ 3.5/5 and ≥3 on three or more individual competencies). This data confirms that the proof-of-concept criterion of the study (i.e., MC competency post-MC training) was met (the efficacy threshold set at n = 4/7, 57% participants was achieved).

### 3.4. Secondary Outcomes

#### 3.4.1. Change in MITI Global Score and Subscales

The mean MITI score of all participants before the MC training was 2.5 ± 0.4 and increased to 3.8 ± 0.4 post-training. Most participants (n = 5/7) reached clinical competency thresholds post-training. In those who did not reached competency, the global MC scores improved (from 2.4 to 3.4/5 in one person and from 2.0 to 3.4/5 in the other). 

Figure 2b illustrates the mean change scores for individual MC competencies before and after the MC training. All competencies improved by more than one point post-training (clinically significant change [29]), except for providing direction which was below the minimal score for competency (i.e., 3.5) before the training. Empathy was the competency with the greatest improvement (2.1 ± 0.7 to 4.1 ± 1.7), whereas providing direction did not improve, likely due to a ceiling effect. Autonomy/support increased and reached minimal competency thresholds (2.4 ± 1.3 to 4.1 ± 0.7). Evocation and collaboration increased but did not reach the competency threshold (1.4 ± 0.8 to 3.1 ± 1.1 and 1.4 ± 0.8 to 2.9 ± 1.1 respectively).

#### 3.4.2. MC Compliance in Clinical Practice

Of the 21 patients who consented to have their consultation recorded or witnessed (Figure 3), 15 had an MC-consistent consultation (i.e., physicians achieving a MITI score of 3.5/5 or greater in that consultation), and 6 had a MC inconsistent consultation. A total of 7/15 of the MC-consistent consultations (47%) were delivered by a single physician. Evocation, autonomy/support and direction all reached minimum competency thresholds of 3.5 or greater on the MITI, whereas collaboration and empathy did not reach minimal competency thresholds (respectively 3.2 ± 0.9 and 3.4 ± 1) (Figure 4). 

#### 3.4.3. Change in Influenza Vaccine Intentions and Vaccination Rates among Patients with RA 

Among the 30 patients initially classified as refusers or hesitant (i.e., regardless of whether the consultation was MC-consistent or not), 18 (60%) reported their intention to receive the vaccine following the consultation. Specifically, among the 13 patients initially identified as vaccine refusers, 7 (54%) were still refusing, 2 (16%) were hesitant and 4 (31%) reported planning to receive the vaccine after their consultation. Among the 17 patients initially identified as vaccine hesitant, after their consultation 1 (6%) indicated that they would refuse the vaccine, 2 (12%) remained hesitant, but the vast majority (n = 14, 82%) reported an intention to receive the vaccine.

Among the 6 patients with a MC-inconsistent consultation, 5 (83%) were initially classified as refusers and 1 (17%) as hesitant. Among the 15 patients with a MC-consistent consultation, 8 (53%) were initially refusers and 7 (47%) hesitant. After a MC-inconsistent consultation, a single patient (1/6, 17%) intended to accept the vaccine as compared to a majority of patients that received a MC-consistent consultation (10/15, 67%, *p* = 0.04) (Figure 5). Further, among patients who received a MC-consistent consultation, 5 (33%) were vaccinated according to provincial health administrative data, compared to none (0%) of the patients in the MC-inconsistent group (*p* = 0.11). 

## 4. Discussion

The results of this proof-of-concept study provide initial evidence suggesting that a 4-h MC training program could increase communication competencies among physicians that may positively impact patient attitudes and behaviors around influenza vaccination. The specific findings indicate that our proof of concept criteria was met, as 5/7 participants (71%) met competency criteria when assessed using a standardized patient with RA who was vaccine-hesitant at one-week post-training. Results also indicated that the majority of physicians were able to successfully use MC in practice, which translated into positive changes in both vaccine intentions and behaviors. In fact, all patients initially vaccine hesitant who got vaccinated received a MC-consistent consultation. 

The results of our proof-of-concept study indicated that some MC competencies were more difficult to acquire than others. For example, collaboration was the competency rated the lowest on the MITI scale both before and after training, whereas providing direction was the competency rated the highest at both time points. In line with the spirit of MC, this highlights the need to shift the traditional ‘provider as expert’ role to a more collaborative one, especially when the focus of the consultation is patient behavior change. Collaborative goal setting for self-management is critical in the context of changing health behaviors like vaccination [15,26,33]. Building collaborative relationships between patients and healthcare providers is also crucial in the context of the COVID-19 crisis, where vaccine uptake is one of the most import prevention behaviours that people can adopt to curb the pandemic. This represents one additional application of MC skills, and further justifies incorporating behavioral change counselling into standard medical practice [34].

In addition to providing important proof-of-concept data, this study also identified methodological challenges that will be important to address in preparation to a randomized controlled trial. One of these challenges is recruiting senior rheumatologists to participate in the training. Despite the fact that many physicians expressed a strong interest in learning how to better address vaccine hesitancy using MC, only one senior rheumatologist attended the training. This could be related to their busier schedule, combined with the lack of incentives offered for attending the training [31]. As such, future trials should consider offering training at convenient times and providing incentives to optimize enrollment and retention (e.g., continuing medical education credits).

Our results are consistent with previous studies that have trained HCPs in MI [23,24]. A recent study reported that a short 2-h MI training could increase physician knowledge about MI techniques, but this increase in knowledge was only observed for up to one 1-month post-training [23]. However, the authors failed to report the extent to which training influenced participants’ actual communication skills in practice. In another study, an 11-h MI-based training program focusing on decreasing vaccine hesitancy among parents in the context of child immunization was delivered to nurses [24]. The training was held on two days separated by 3-months. The results showed an increase in MI knowledge and skills (e.g., asking open-ended questions) immediately after the first day of training. However, only the knowledge effect persisted over the 3-month follow-up period. Similarly, a systematic review of the impact of MI training on physicians’ skills and patient outcomes related to substance abuse [35] found that among the 12 studies that assessed physicians’ MI skills at 20 weeks post-training, none of these studies reported that participants maintained MI competency at follow-up according to the MITI global score. Maintaining MC skills competency over time is an important challenge that suggests the need for continued supervision and/or booster sessions to reinforce learning. This is supported by a recent meta-analysis showing greater effects on participants’ MI skills when the training included booster sessions [36]. There is also evidence that providing physicians with periodic individual feedback about their MC techniques might also promote the maintenance of competences over time [37,38,39].

### Limitations and Strengths 

This study has several limitations. The data generated in this study was primarily from rheumatology residents (with a single exception) thus the results may not apply to more senior rheumatologists who are expected to embrace a mentor/teaching approach rather than in a learning one [40,41]. Caution should also be used before generalizing the results to other type of vaccine, such as the new mRNA COVID-19 vaccine. With regards to the evaluation tool, MITI 3.0 was designed to assess competency in MI, not MC, and was validated for use during a 20-min consultation. In this study, we used the MITI during brief consultations lasting up to 10 min. This highlights the need to develop new MC competency assessment tools better adapted for brief consultations that better fit clinical practice. We are currently validating such a tool [27]. Another limitation was that nine out of 30 consultations did not have fidelity assessments due to the absence of patient consent. Although these 9 patients were classified as hesitant (as opposed of refusers), it is still possible that patients with more negative vaccine attitudes were less likely to consent to have their consultation witnessed or recorded, which may have resulted in a bias towards observing more positive vaccine outcomes. Given the objectives of this proof-of-concept study were to obtain insight about the preliminary “signal strength” of our training program, the results should be interpreted as evidence of efficacy, which would require a randomized design.

Despite these limitations, this study has a number of important strengths. First, the MC training was developed based on established theories of behavior change using specific components of behavior intervention. It was tailored to the target population (i.e., rheumatology physicians) and a specific health behavior (i.e., influenza vaccination among patients with vaccine hesitancy). Fidelity was assessed ensuring that all MC modules and intervention components were delivered as intended. Two data collection methods (i.e., a standardized patient and actual patients) were used allowing to standardize the competency assessment for the primary outcome and assess skill competency in “real life”. Finally, the competency of use of MC by training participants, both with standardized and real patients, was evaluated using an objective assessment (MITI) by an observer rather than self-reported by the participants themselves.

## 5. Conclusions

In this proof-of-concept study, a short MC training program improved communication competencies of rheumatology physicians as well as influenza vaccine intentions in patients targeted by the intervention. Future studies should consider using incentives to increase physician participation, strengthening aspects of the training program (e.g., collaboration competencies) and providing booster sessions to enhance the long-term maintenance of skills. The positive results of this proof-of-concept study suggest that the short MC training program, incorporating some modifications, can move forward to be tested in a randomized controlled trial. To our knowledge, this is the first study documenting the impact of a brief MC training program on physicians’ communication competencies that translated into positive vaccine intentions and behaviors among patients being vaccine hesitant. 

## Figures and Tables

**Figure 1 vaccines-10-00143-f001:**
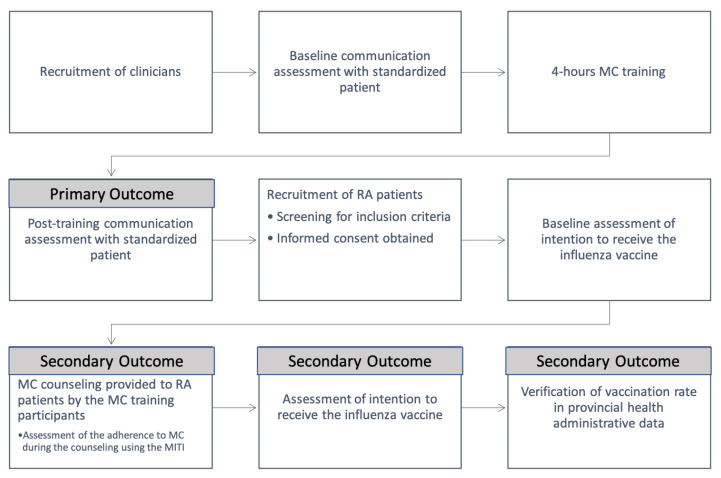
Overview of study design. MC = motivational communication; RA = rheumatoid arthritis; MITI = motivational interviewing treatment integrity.

**Figure 2 vaccines-10-00143-f002:**
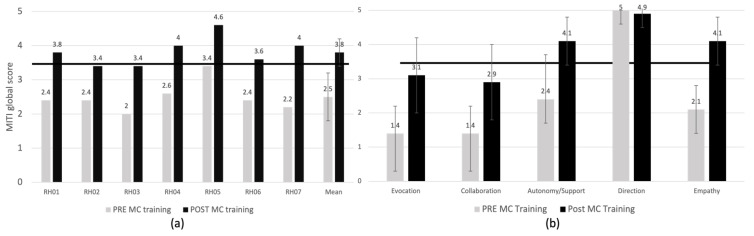
(**a**) MITI global score of physicians during consultations with a standardized patient before and after MC training. MITI = motivational interviewing treatment integrity scale; MC = motivational communication; RHXX indicates the MC training participant ID. Scale corresponds to: 1 = Low presence of the competency during the discussion; 5 = High presence of the competency during the discussion. 3.5 corresponds to minimal score to be considered MI/MC competent. Minimal competency threshold is indicated by a bold line at global score of 3.5. (**b**) MITI sub-scores by MC competency. Scale corresponds to: 1 = Low presence of the competency during the discussion; 5 = High presence of the competency during the discussion. 3.5 corresponds to minimal score to be considered MI/MC competent. MITI: Motivational Interviewing Treatment Integrity scale. Minimal competency threshold is indicated by a bold line at global score of 3.5.

**Figure 3 vaccines-10-00143-f003:**
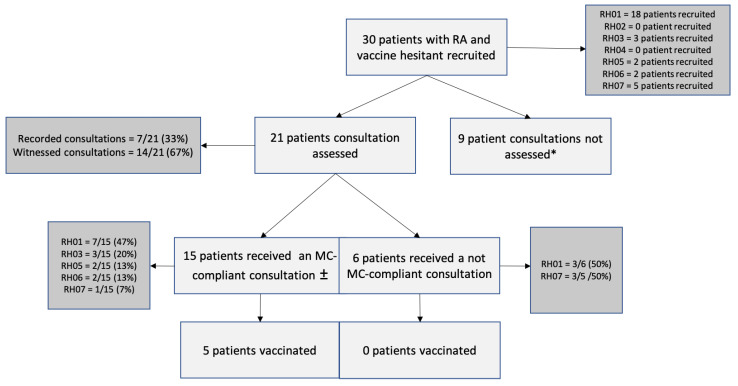
Flow chart of recruitment of patients with RA being vaccine hesitant who underwent a consultation with a MC-trained participant. RA = rheumatoid arthritis; MC = motivational communication, RHXX indicates the MC training participant ID. * Patients did not provide consent to have their MC counselling session recorded. ±Counselling session reached minimal global score of 3.5 on the Motivational Interviewing Treatment Integrity (MITI) 3.0 and reached 3/5 on at least 3 individual competencies.

**Figure 4 vaccines-10-00143-f004:**
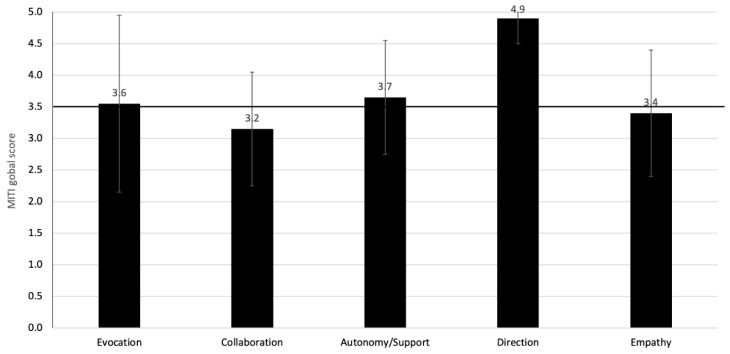
MITI scores by motivational communication competency expressed during recorded/witnessed consultations with patients after receiving the training (n = 21). Note: Scale corresponds to: 1 = Low presence of the competency during the discussion; 5 = High presence of the competency during the discussion. 3.5 corresponds to minimal score to be considered competent in motivational communication. MITI = motivational interviewing treatment integrity scale. Minimal competency threshold is indicated by a bold line at global score of 3.5.

**Figure 5 vaccines-10-00143-f005:**
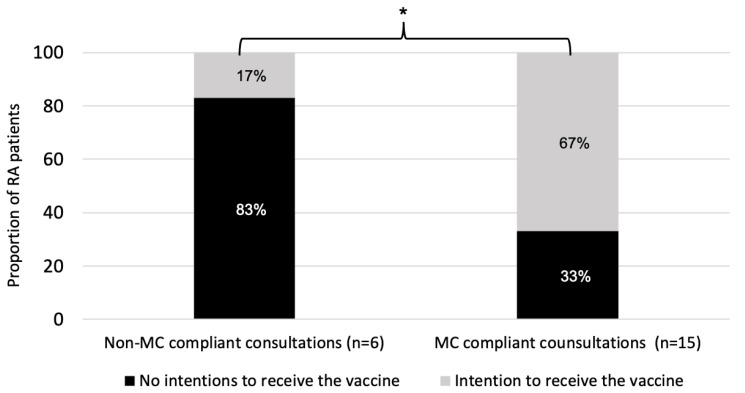
Patients’ intentions to receive the vaccine after undergoing a consultation with the physician. RA = rheumatoid arthritis; MC = motivational communication. * *p* = 0.04.

## Data Availability

The data presented in this study are available on request from the corresponding author.

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
