# Peer review of "Training Physicians in Motivational Communication to Address Influenza Vaccine Hesitation: A Proof-of-Concept Study"

_vaccines, 2022, doi:10.3390/vaccines10020143_

Round 1

Reviewer 1 Report

It is a preliminary study and provides a low level of evidence.

There is no control group in the study and no way to control confounding factors and biases.

Selection bias in participated rheumatologist is obvious.

According to the almost half of the participants consented to be observed not recorded their consultations, measurement bias in assessment of pre- and post-interventions are inevitable.

Learning curve bias for consultants specifically one with much more consultations in the study, is another source of bias.

Appropriate analysis for numerical outcome variables has not been performed and has not been reported.

Because of patients who did not provide consent to have their MC counselling session recorded shall be considered as missing, a sensitivity analysis shall be performed for better understanding the effect of these patients.

Decision: Major revision, Improvement of the analysis and more discussing about the limitations of the study are essential.

Author Response

Reviewer 1

Comments and Suggestions for Authors

  • It is a preliminary study and provides a low level of evidence. There is no control group in the study and no way to control confounding factors and biases.

Response: We appreciate the reviewer’s concerns, and acknowledge that publishing these early-phase development trials is somewhat of a new concept for many reviewers. While it is true that this study does not have a control group, this is actually inappropriate for proof of concept trials (whose objective is NOT to test the efficacy of an intervention in a randomized study, but rather, the extent to which an intervention shows a ‘signal’ of being able to achieve a pre-specified minimum threshold of impact, often defined as reaching a clinically significant threshold rather than a statistically significant difference between two groups). The reason for doing these early phase studies (which mirror the development process of pharmaceuticals), it to ensure that by the time we conduct a phase III fully powered RCT, that we are confident that our intervention is ‘strong enough’ to produce the desired impact. Failing to conduct these preliminary studies would potentially waste valuable resources and would be unethical from a participant perspective, by subjecting them to a trial that has uncertain chances of being efficacious. We have added some additional clarification of the objective of this study and rationale for the design on page 3, line 105-111. This study (and design) is aligned with recommendations by experts on how to develop and test behavioural interventions, as described by the ORBIT Model, and we invite the reviewers to consult this document for further information: Czajkowski et al., 2015; Craig et al., 2008.

  • Selection bias in participated rheumatologist is obvious.

Response: We agree with the reviewer that due to its design, selection bias is a potential limitation. However, the goal of the proof of concept study is not be representative or generalizable, it is to get a preliminary sense of the ‘strength’ of particular intervention in the target population. Nonetheless, we have discussed this as a consideration in the limitations section (line 451-458).

  • According to the almost half of the participants consented to be observed not recorded their consultations, measurement bias in assessment of pre- and post-interventions are inevitable.

Response: We appreciate the reviewer’s comment. As a point of clarification, we were able to assess the extent to which physicians conducted a MC-consistent or inconsistent consultation in 21 out of 30 patients, either by viewing a recording of the consultation, or by completing the assessment ‘live’ while observing the consultation in real time. This indicates that we were only lacking data in 30% (not 50%) of patients. While we acknowledge this as a limitation, we are confident that we had enough data (70% of participants consented to being recorded or observed) upon which to assess treatment fidelity among physicians. This limitation is discussed on page 12, lines 451-458.

  • Learning curve bias for consultants specifically one with much more consultations in the study, is another source of bias.

Response: We thank the reviewer for pointing this out and acknowledge that indeed, we had one physician in the study who conducted more consultations than others. However, we were limited to assessing all of the vaccine consultations that occurred among the physicians we enrolled in the study, and one ended up doing more of these consultations than others, and we did not anticipate that in advance. However, it should be noted that every patient can bring different vaccination concerns and barriers to a consultation, so physicians may be required to pull upon different communication skills to engage patients in a conversation about vaccination. Therefore, we consider each individual consultation as a unit of measurement. Nonetheless, we think this is an important point to consider when interpreting the data, and have provided additional information about this in the limitations section of the manuscript (see page 12, line 442-444).

  • Appropriate analysis for numerical outcome variables has not been performed and has not been reported.

Response: We thank the reviewer for raising this issue, and would like to take a moment to make an additional point of clarification regarding the design and outcome measurements in this study. As mentioned above, this is a proof of concept study whose main objective is to provide evidence of the extent to which an intervention shows a ‘signal’ of being able to achieve a pre-specified minimum threshold of impact, often defined as reaching a clinically significant threshold rather than a statistically significant difference between two groups. While the appropriate statistical analysis strategy for the latter (efficacy testing) is testing for statistically significant differences between groups in an appropriately powered RCT, the appropriate analytical strategy for a proof of concept study is typically the proportion of participants achieving a clinically significant threshold of change on the primary outcome (in our case, achieving minimal competency levels on the MITI). We a-priori set our ‘success’ threshold criterion at greater than 50% (ie, at least 4 out of 7 of our participants had to achieve a minimum global competency of 3.5/5 on the MITI, which is what the MITI defines as the competency threshold). So rather than test for statistically significant differences between 2 groups, we are actually simply reporting whether or not we met our criteria (yes, no) – i.e., did at least 50% (or at least 4/7 physicians) achieve scores of 3.5/5 or greater on the MITI), and in our study, the answer is yes: 5/7 achieved this criterion, which led us to concluded that we have satisfied our proof of concept. We have provided further clarifications of the design of the study (see page 3, lines 106-110 and page 7, lines 256-259) and further justification for the analysis strategy on page xx, lines 233-235. As a reminder, these methods are aligned with expert recommendations as outlined in the ORBIT Model (Czajkowski et al., 2015; Craig et al., 2008).

  • Because of patients who did not provide consent to have their MC counselling session recorded shall be considered as missing, a sensitivity analysis shall be performed for better understanding the effect of these patients.

Response: While we appreciate the reviewer’s comment, given that the vast majority (70%) of the participants provided consent for recording or observation, the small sample size, and the overall objective of the study (which is not efficacy but detecting a ‘signal’ for the strength of the intervention), we don’t believe it would appropriate or greatly informative to conduct a sensitivity analysis (though we agree that this WOULD be completely appropriate in the context of a fully powered randomized controlled trial, in the event there were a significant number of refusals, drop-outs or non-responders). A description of these 9 patients has been added (they were initially all hesitant, lines 278-279). We also mentioned this potential source of bias in the limitation section (lines 451-455). It,  We hope the reviewers find this response satisfactory.

References:

Craig, P., Dieppe, P., Macintyre, S., Michie, S., Nazareth, I., & Petticrew, M. (2008). Developing and evaluating complex interventions: the new Medical Research Council guidance. BMJ, 337, a1655. doi: 10.1136/bmj.a1655

Czajkowski, S. M., Powell, L. H., Adler, N., Naar-King, S., Reynolds, K. D., Hunter, C. M., . . . Charlson, M. E. (2015). From ideas to efficacy: The ORBIT model for developing behavioral treatments for chronic diseases. Health Psychol, 34(10), 971-982. doi: 10.1037/hea0000161

Reviewer 2 Report

This study seems very sound.  It shows that a training course in Motivational Communication is able to allow health providers the ability to change the behavior of vaccine hesitant patients. 

I have one concern about a control group.  I would like to know how many vaccine hesitant patients change their mind after being seen by providers who have not been given the Motivational Communication training. Also only 5/15 patients actually received the vaccine.  

29 November (mail):

--What does it add to the subject area compared with other published
material?

It offers insight into the motivational communication method.

--Are the conclusions consistent with the evidence and arguments
presented?

If not, how improve. It does however I think they can address a better control group. How do normal patient visits (without MC) change patient decisions. 

--Are there any methodological limitations in this research? Such as...

Those were addressed.  As mentioned they did discuss that patients who refuse vaccination may also refuse monitoring. Those are part of it. I also believe the idea of the better group may be a limitation. I hope they can address that. 

--How is the structure of the article?

Structure is fine. 

--Are references correct and necessary?

Yes

--Are there any English grammar mistakes?

No

--Has materials been provided completely?

I wish some more detail on Motivational Communication can be added to make it easier  

In any case the article seems publishable.  These suggestions can improve upon it. No "experiments" need doing but just address the limitations and add more detail on Motivational Communication making it clearer.

Author Response

Reviewer 2

Comments and Suggestions for Authors

This study seems very sound.  It shows that a training course in Motivational Communication is able to allow health providers the ability to change the behavior of vaccine hesitant patients. 

I have one concern about a control group.  I would like to know how many vaccine hesitant patients change their mind after being seen by providers who have not been given the Motivational Communication training. Also only 5/15 patients actually received the vaccine.  

Response: This is indeed an interesting question! Since study participants were only physicians who received MC training (and who consenting to be included in the study), we did not collect (and could not collect) data on physicians who did not receive the training, or their patients. We agree with the reviewer that this is an interesting and important question, and one that would be more appropriately answered within the context of a RCT (which was not the purpose nor design of this study). So although our study was not designed to answer this question, we do have data among vaccine-hesitant patients who did not receive an MC-consistent consultation: none of them changed their mind and received the vaccine (see page 10, line 368-370 and Figure 3). 

Reviewer 3 Report

Through this paper. the authors aim to discuss their 4-hour Motivational Communication (MC) training program in order to help physicians to address hesitancy related to influenza vaccination among patients living with rheumatoid arthritis.

Although this paper could be of interest to the readers, I suggest several modifications in order to improve it.

All terms should be inserted in the extension form (i.e. Randomized Controlled Trials (RCTs), line 69).

The introduction section should be shortened: please insert only the information that is functional to present the aims of the study. Moreover, the study's aims should be clearly described.

The "material and methods" and "results" sections reported all useful information.

The discussion section should be improved: the authors missed to compare their data to international data. Moreover, it could be of interest to insert a short paragraph about the COVID-19 hesitancy in order to increase the interest of the article, proposing a comparison. Finally, in the limitations, the authors should insert the low number of physicians as well as the low number of patients.

Author Response

Reviewer 3

Comments and Suggestions for Authors

Through this paper. the authors aim to discuss their 4-hour Motivational Communication (MC) training program in order to help physicians to address hesitancy related to influenza vaccination among patients living with rheumatoid arthritis.

Although this paper could be of interest to the readers, I suggest several modifications in order to improve it.

All terms should be inserted in the extension form (i.e. Randomized Controlled Trials (RCTs), line 69).

Response: Done

The introduction section should be shortened: please insert only the information that is functional to present the aims of the study. Moreover, the study's aims should be clearly described.

Response: We thank the reviewer for his comment. We have shortened the introduction. We also provided additional details about the objectives of this study (lines 107-110).

The "material and methods" and "results" sections reported all useful information.

The discussion section should be improved: the authors missed to compare their data to international data. Moreover, it could be of interest to insert a short paragraph about the COVID-19 hesitancy in order to increase the interest of the article, proposing a comparison.

Response: We added a concise discussion about the COVID-19 vaccine (page 11, lines 393-397).

Finally, in the limitations, the authors should insert the low number of physicians as well as the low number of patients.

Response: We added details about the limits of this proof-of-concept study in the limitation section (lines 455-458).

Round 2

Reviewer 1 Report

According to the explanations of the authors and major source of bias in the study and low scientific soundness,  Decision is reject

Only in a letter format it may be considered for publication as a preliminary report.

25 December 2021 additional comments by mail:

As a short communication, it can be acceptable for publication.

Author Response

We would like to publish this manuscript as a paper for several reasons. While we acknowledge the concerns of the reviewer (which seem primarily related to sample size and selection bias), it is our belief that they have reviewed this paper as if it were a small efficacy trial, which it is not. It is a proof of concept study where small sample sizes (in this case, 7 physicians and 30 patients) are not only the norm but recommended (see Czajkowski et al, Health Psychology, 2015), to determine early in the intervention development process, the extent to which an intervention (in this case, training program) is 'strong enough' to produce the intended effects on the outcomes (in this case, competency in MC skills). We have presented a carefully developed intervention and compelling evidence that it demonstrates strong potential for helping physicians achieve competency in motivational communication, but also, that when physicians use MC with patients, patients who were hesitant accept to be vaccinated.   

That being said, we believe that reducing this paper down to a letter that would focus on the results rather than the methods and the types of outcomes that need to be measured (which are, in our view, are the most important aspect to publish), would not do this work justice, or make as important a contribution. One of the major gaps in the behavioural intervention literature is the failure to carefully develop and pre-test interventions prior to moving to complex and costly phase III RCT's. Our goal here is not only to showcase a highly promising training program for physicians that could help them better support patients in their decision-making around vaccination (a highly relevant topic in the context of the COVID-19 pandemic), but also to illustrate how these interventions should be developed and tested. There are few published articles of this early-phase work, which we think perpetuates the impression that it is not of value or publishable. As co-Lead of the International Behavioural Trials Network (www.ibtnetwork.org), and an international expert in this field, I (and the IBTN) are committed to changing this, and scientists in the field need to see how this is done and that it is valued by the scientific community.